# Effect of Magnesium Substitution on Structural Features and Properties of Hydroxyapatite

**DOI:** 10.3390/ma16175945

**Published:** 2023-08-30

**Authors:** Vladimir S. Bystrov, Ekaterina V. Paramonova, Leon A. Avakyan, Natalya V. Eremina, Svetlana V. Makarova, Natalia V. Bulina

**Affiliations:** 1Institute of Mathematical Problems of Biology—Branch of Keldysh Institute of Applied Mathematics, Russian Academy of Sciences, 142290 Pushchino, Russia; ekatp11@gmail.com; 2Physics Faculty, Southern Federal University, 344090 Rostov-on-Don, Russia; laavakyan@sfdeu.ru; 3Institute of Solid State Chemistry and Mechanochemistry, Siberian Branch, Russian Academy of Sciences, 630128 Novosibirsk, Russia; eremina@solid.nsc.ru (N.V.E.); makarova@solid.nsc.ru (S.V.M.); bulina@solid.nsc.ru (N.V.B.)

**Keywords:** hydroxyapatite, magnesium substitution, density functional theory, lattice parameters, formation energy, mechanochemical synthesis, IR spectrum, bulk modulus, band gap

## Abstract

Hydroxyapatite (HAP) is the main mineral component of bones and teeth. It is widely used in medicine as a bone filler and coating for implants to promote new bone growth. Ion substitutions into the HAP structure highly affect its properties. One of the most important substituents is magnesium. This paper presents new results obtained using high-precision hybrid density functional theory calculations for Mg/Ca substitutions in HAP in a wide magnesium concentration range within a 2 × 2 × 2 supercell model. Experimental data on the mechanochemical synthesis of HAP-Mg samples with different Mg concentrations are also presented. A comparison between the experiment and the theory showed good agreement: the HAP-Mg unit cell parameters and volume decreased with increasing degree of Mg/Ca substitution. The changes in the distances between the Ca and O, Ca and H, and Mg and O ions upon Mg/Ca substitution in different calcium positions was analyzed. The resulting asymmetry and distortion of the cell parameters were evaluated. It was shown that bulk modulus, energy levels, and band gap depend on the degree of Mg substitutions in the Ca1 and Ca2 positions. The formation energies of Mg/Ca substitutions showed non-monotonic behavior that was different for Ca1 and Ca2 positions. The Ca2 position had a slightly higher probability (~5 meV/f.u.) of substitution than Ca1 position at a Mg concentration x = 0.5. At x = 1, substitution in both positions can coexist. The simulated IR spectra for different Mg/Ca substitutions showed that Mg in the Ca2 position changes the IR spectrum more significantly than Mg in the Ca1 position. Similar changes were recorded in the IR spectra of the synthesized samples. The electronic structure is shown to be sensitive to the number and position of substitutions, which may be used to tweak the optical properties of the HAP-Mg material.

## 1. Introduction

Materials based on calcium phosphates are important substances that are actively used in medicine, biology, ecology, and catalysis [1,2,3,4]. One of them is hydroxyapatite (Ca_10_(PO_4_)_6_(OH)_2_, HAP), which is traditionally used in medicine and other areas such as environmental purification, photocatalysis, photoluminescence, cell imaging, drug delivery, and cancer treatment [5,6,7,8,9,10,11,12].

HAP is primarily distinguished as the main mineral inorganic component of the bones (up to 50% of bone mass) and teeth (up to 96% in enamel) of vertebrates [3,13]. HAP crystallizes in the form of platelet nanocrystals (2–3 nm thick and ~25–45 nm in size) in the spaces between tropocollagen fibrils, along with the organic component (collagen) and living bone cells (osteoclasts, osteoblasts, and osteocytes) forming and strengthening the bone structure [14,15]. Due to such innate biological activity and biocompatibility, synthetic HAP is the most widely used biomaterial in various medical applications in bone and dental surgery as a filler and coating for implants [3,4,5,6,7,8,9]. The biological HAP differs from synthetic analogs in its stoichiometric imbalance and the presence of a significant amount of various impurities of ions (F^−^, Cl^−^, Mg^2+^, Na^+^, K^+^, Fe^2+^, Mn^2+^, Zn^2+^, Sr^2+^, etc.) and molecular ionic groups (CO_3_^2−^, SiO_4_^4−^) [7,16,17]. The concentration of these impurities does not exceed 3–5%, but they play an important role and determine the biological, mechanical, and also, in the case of enamel, optical properties of biogenic HAP.

The direct use of HAP for implant production is limited due to the difference in mechanical properties between stoichiometric HAP and native bone. For example, it is known that fracture toughness (K_Ic_) is a decisive material property in the development of reliable grafts for bearing loads [17]. Stoichiometric HAP turns out to be a rather fragile material, since its K_Ic_ value is in the range of 0.5–1 MPa·m^1/2^ [18]. This is noticeably lower than that of human cortical bone, which provides K_Ic_ characteristics in the range of 2–12 MPa·m^1/2^ [18,19]. In this regard, bone implants are made of the most suitable material (usually titanium) with HAP coating [9], which provides better adhesion and biocompatibility with emerging living bone tissue, as well as the reproduction of living bone cells [8,13].

The crystal structure of apatites is quite flexible and makes it possible to form many different compositions with the incorporation of impurity ions into the crystal lattice. The mechanical, biological, and physicochemical properties of such HAPs are closely related to changes in both the atomic-molecular and the electronic structure of HAP caused by substitutions for impurity ions and ion groups. Therefore, it is very important to understand the relationship between structural aspects and material properties when various substitutions or insertions are included in the HAP structure [4,5,6,7,8].

One of the effective ways to study HAP structural changes is computer modeling, especially with the use of the modern methods of density functional theory (DFT) [20,21,22,23,24]. In particular, our calculations showed that the presence of various structural defects in HAP, such as oxygen vacancies, OH vacancies, and substitutions, affect its properties [24,25,26,27,28,29,30].

A number of works [10,24,25,26,27] including precision measurements at the DESY synchrotron [31,32] show that the band gap of an ideal stoichiometric HAP is in the rage of E_g_ = 7.4–7.7 eV, while for HAP with defects such as oxygen and OH vacancies or substitutions, it is much lower E_g_ = 3.4–4.3 eV. These results, which coincide with the data obtained in our DFT calculations, show the reliability and validity of the developed theoretical DFT approaches and their applicability in problems of numerical studies of HAP with various structural defects and substitutions. The features of such non-stoichiometric HAP can be more efficiently studied by combining experimental and theoretical methods particularly with a combination calculating HAP models and adjusting the modes and conditions of synthesis [33,34], taking into account high “flexibility” of the HAP crystal lattice and its high structural stability [17,35,36,37].

When considering cation substituents, magnesium turns out to be one of the most important divalent ions associated with biological HAP [7,38,39]. Enamel, dentin, and bones contain 0.44, 1.23, and 0.72 wt.% of Mg, respectively [39].

A lack of magnesium makes the bone more fragile and causes impaired osteogenesis [40,41,42]. The review [17] notes that a higher value of the strength index (fracture toughness) of K_Ic_~2.7 MPa·m^1/2^ was achieved by alloying HAP with magnesium at a concentration of 0.6 wt.% of Mg [43].

Experimental studies of the effect of magnesium at various concentrations on the properties of HAP are based on the preparation of Mg-substituted HAP (HAP-Mg) samples using various methods and the subsequent characterization of their properties using different techniques. HAP-Mg is usually obtained using precipitation and hydrolysis methods. The maximum amount of Mg^2+^ replacing Ca^2+^ varies in different publications [39,44,45,46]. Thus, using the mechanochemical–hydrothermal technique, Riman et al. [47] reported stable-phase pure-magnesium-substituted crystalline HAP containing from 2.0 to 29 wt.% of Mg, where not less than 75 wt.% Mg was substituted for calcium ions in the HAP lattice structure [39]. The complete replacement of Ca^2+^ with Mg^2+^ in the apatite synthesized in a high pH solution was also reported [39,48]. Similarly, in [39], the wet chemical-precipitation method was used to prepare HAP-Mg nanocrystals [49]. Various other methods are also used to prepare HAP compositions and coatings with the introduction of magnesium (and other elements), in particular, based on mechanochemical synthesis [50,51,52]. Studies are also being carried out to obtain coatings based on HAP doped with various concentrations of Mg on a nanostructured titanium surface using electrochemical methods [53]. Recently, experimental methods have also been developed for applying HAP coatings with additionally introduced elements (Mg, Sr, etc.) from a sputtered target using magnetron sputtering methods [54].

Various computational approaches have been used to study the effect of Mg substitution on the structural characteristics of HAP.

Advance computational approaches such as DFT calculations have been increasingly used to study the effect of magnesium in HAP [39,55,56,57,58].

However, the data obtained are still quite contradictory and not unambiguous. For example, different values of formation energy for Mg/Ca substitution and different preferred positions for the substitution were obtained in the works [39,54,55,56,57,58]. Shortcomings in the used models, applied methods, and approaches of DFT were perhaps the reason for the different obtained data. Therefore, further extensive studies are required to clarify the effects of Mg substitutions on HAP’s structure and its properties.

The aim of our work is to study the effect of magnesium on the structural features and properties of HAP both with theoretical methods using accurate high-precision DFT methods applying new hybrid functions for modeling and with experimental ones using a mechanochemical method of synthesis and different analytical equipment. The obtained experimental and theoretical data are compared for various concentrations of magnesium cations replacing calcium cations. These studies are very important for bone-tissue engineering. The results obtained here will contribute to better design of conditions for increasing the mechanical strength and biocompatibility of magnesium-modified HAP.

## 2. Materials, Models and Methods

### 2.1. Experimental Part

Mechanochemical synthesis of HAP-Mg samples was carried out using a mechanochemical method in an AGO-2 planetary mill. The synthesis was carried out in water-cooled steel drums with steel balls. Detailed information concerning the conditions of synthesis can be found in refs. [33,50,51,52]. The initial components for the synthesis of HAP-Mg were anhydrous calcium hydroorthophosphate CaHPO_4_ (pure grade), calcined calcium oxide CaO (pure grade), and magnesium dihydrogen phosphate dihydrate Mg(H_2_PO_4_)_2_·2H_2_O (pure grade). The duration of processing of the initial mixture in the mill was 30 min. The initial components were used in a stoichiometric ratio based on the assumption that calcium cations are replaced by magnesium cations in accordance with the reaction
(6 − 2x)CaHPO_4_ + (4 + x)CaO + xMg(H_2_PO_4_)_2_·2H_2_O → Ca_10−x_Mg_x_(PO_4_)_6_(OH)_2_ + nH_2_O(1)
where x = 0, 0.25, 0.5, 1.0, 1.5, 2.0.

Samples are labeled as “xMg”, where x corresponds to the number of substituted calcium atoms in the chemical formula of HAP (in this case, there is one unit cell of hydroxyapatite).

Synthesized materials were characterized using X-ray diffraction (XRD) and Fourier transmission infrared (FTIR) spectroscopy. A D8 Advance diffractometer (Bruker, Karlsruhe, Germany) with CuK–radiation was utilized in the Bragg–Brentano geometry for recording XRD patterns. The patterns were analyzed with the help of the ICDD PDF-4 powder diffraction database (2011). The refinement of unit cell parameters was carried out using the Rietveld method in the Topas 4.2 software (Bruker, Karlsruhe, Germany). FTIR spectra were recorded on a Scimitar-FTS 2000 spectrometer (Digilab LLC, Randolph, MA, USA) using the KBr pellet method. An energy-dispersive X-ray analyzer Noran System 7 (Thermo Fisher Scientific, Waltham, MA, USA) attached to a TM-3400N Tabletop scanning electron microscope (Hitachi, Tokyo, Japan) was used for the elemental analysis of the synthesized samples.

### 2.2. Computational Part

#### 2.2.1. Methodological Features

In this work, we apply the computational approach developed by us earlier, which allows us to obtain high-precision results by performing DFT calculations in 2 stages. This approach was successfully tested and used in [24,25,26,27,28] for DFT calculations on the HAP models of one unit cell and a supercell (Figure 1). It is based on the Purdue, Burke, and Ernzerhof (PBE) functional in the generalized gradient approximation (GGA) [59] in combination with the Heyd, Scuseria, and Ernzerhof (HSE06) [60,61] hybrid exchange-correlation functional.

#### 2.2.2. Computational Details and Parameters

The electronic structure of the ground state of considered atomic models was calculated using the QUANTUM ESPRESSO package [62] in the framework of the DFT approach with the above-mentioned functionals. The core electron states are described using optimized norm-conserved (ONCV) DFT pseudopotentials [63,64], while the kinetic energy cut-offs were set to 60 and 240 Ry for the Khon–Sham wave function expansions and semi-local potential, correspondingly. The exact exchange operator was calculated using the reduced cut-off energy of 120 Ry. The forces acting on the atoms were estimated based on the calculated electronic structure of the ground state. The search for the stable atomic configuration was carried out using an iterative method for the numerical optimization of the total energy, which was continued until any component of the force at any atom was higher than 10^−4^ Ryd/Bohr, and the energy difference was 7 × 10^−6^ Ry = 10^−4^ eV. It should be noted that the application of hybrid functionals, in particular HSE06, considerably improved the predicted value of the band gap E_g_ of dielectric materials, including HAP ceramics [26].

The calculations of phonons required for the modeling of the IR spectra were performed within the PBE approximation. The IR transmission *T* was evaluated as *T* = exp(−K·σ), where *K* is the parameter that depends on the density and depth of the sample (we took K = 0.001), and σ values are the calculated IR intensities, which were smeared with a Gaussian function of 1 meV in width.

#### 2.2.3. Basic Structure Models

In this work, we consider substitutions associated with magnesium atoms/ions Mg/Ca. To construct and study the models of HAP structures with substitutions having the initial chemical formula Ca_10_(PO_4_)_6_(OH)_2_ and with Mg/Ca substitutions Ca_10−x_Mg_x_(PO_4_)_6_(OH)_2_ with Mg/Ca substitutions (Equation (1)), we started from the HAP unit hexagonal cell (*P*6_3_ group) consisting of 44 atoms (10 Ca atoms, 6 PO_4_ groups, and 2 OH groups) [16,25] (Figure 1).

Next, the HAP 2 × 2 × 2 orthogonal supercell (containing 352 atoms) was used for numerical studies. For the clarity of presentation, the structures on the illustrations were shifted to demonstrate two OH channels including surrounding calcium atoms with different crystallographic positions, Ca1 and Ca2. The cell parameters for the supercell and for the hexagonal unit cell of the initial HAP model were obtained as a result of optimization [12,24,27]: *a = b* = 18.962 Å/2 = 9.481 Å and *c* = 13.717 Å/2 = 6.8585 Å. An important feature of the HAP structure is that the unit cell of HAP contains 10 Ca cations located in two nonequivalent positions: four cations are in the Ca1 position, and six cations are in the Ca2 position [5]. Accordingly, there are 80 Ca in the supercell, including 32 Ca cations in the Ca1 position and 48 Ca cations in the Ca2 position (see Figure 1).

We denote the *degree of substitution* based on the chemical reaction (1) in the HAP unit cell as follows: Ca_10−x_Mg_x_(PO_4_)_6_(OH)_2_. This means that for x = 1, only 1 Ca atom of 10 calcium atoms is replaced by 1 Mg atom. For the 2 × 2 × 2 supercell consisting of 8 elementary cells, these values should be multiplied by 8.

The substituted HAP-Mg structure is modeled by replacing Ca atoms with Mg atoms in the Ca1 and Ca2 positions with different numbers of substituting Mg atoms: nMg/Ca1, nMg/Ca2 with *n* = 1, 2, 4, 8, 12, 16 in the HAP supercell containing 80 Ca positions.

The selected positions of Ca atoms for substitution by Mg are given in Appendix A and are shown in Appendix A.

The magnesium concentration *x* in reaction (1) is related to the number of substitutions *n* as x = *n*/8, where 8 is the ratio of calcium positions in the supercell and in the unit cell: 80/10 = 8 (equal to the number of unit cells in a supercell). For example, *n* = 1 corresponds to x = 0.125, and the highest considered degree of substitution *n* = 16 corresponds to x = 2.

## 3. Results and Discussion

### 3.1. Experimental Results

Figure 2 shows XRD patterns of the powders synthesized with the addition of different amounts of magnesium. In the diffraction patterns, there are reflections characteristic of the HAP phase, corresponding to card PDF 01-76-0694, indicating that all synthesized samples are single-phase. From Table 1, it can be concluded that the lattice parameters of HAP in the HAP-Mg samples, as well as the cell volume, decrease with the increasing amount of introduced magnesium. Such a dynamic can be explained by the influence of the smaller ion radius of magnesium upon calcium substitution. The substituent also reduces the crystallinity of the material in question (Table 1). In the lattice, the substituent ions represent point defects that complicate crystallite growth.

It is worth noting that in spite of the large number of publications devoted to the study of the properties of magnesium-substituted apatites, there is no unambiguous X-ray evidence that magnesium cations replace calcium cations in the studied samples without applying heat treatment during synthesis. The available publications do not provide lattice parameters [65,66,67] the parameter dynamics is absent [68], or there are illogical fluctuations/changes in parameters [69]. Our data are in agreement with only one work [39], which shows a pronounced decrease in the unit cell parameters and its volume with increasing concentrations of injected magnesium.

The EDX microanalysis (Appendix A) of synthesized powders (Appendix A) showed that with an increase in the concentration of magnesium introduced, the concentration of magnesium in the samples increased. The concentrations of detected elements are close to the expected ones. As the XRD data (Figure 2a) show only the apatite phase in all samples, the observed concentration of magnesium corresponds to magnesium in the apatite structure.

FTIR spectra of the synthesized samples are present in Figure 3. The spectrum of unsubstituted HAP contains absorption bands that are characteristic of the HAP structure: bands of the phosphate ion group (570, 602, 962, 1048, and 1088 cm^−1^) and of the OH group (630 and 3570 cm^−1^). In the spectrum, the wide bands of carbonate ions (at 1420 and 1470 cm^−1^) and sorbed water (1630 and 3420 cm^−1^) are also observed. As concern the spectra of substituted samples, the phosphate ion bands persist, widening with increasing magnesium concentration, while the hydroxyl group bands decrease until they completely disappear. This behavior of the hydroxyl group band may indicate a decrease in the number of hydroxyl groups, which is inconsistent with the substitution mechanism according to reaction (1).

According to our previous investigations [33], a small amount of water released during mechanochemical synthesis, which proceeds according to reaction (1), is predominantly adsorbed by apatite particles. A small amount of water is embedded in the crystal lattice. Adsorbed and lattice water is completely evaporated from particles after heating to 500 °C without any structural transformations. The lattice water may localize in the OH channel [5] and be the reason for a decrease in the number of hydroxyl groups. To avoid the influence of water, we carried out a thermal treatment of synthesized samples at 500 °C.

Figure 2b confirms that this thermal treatment did not affect the phase composition of the synthesized samples while the values of lattice parameters changed (Table 1), but the overall dynamic of the lattice contraction as the concentration of introduced magnesium increased remained the same. At the same time, some changes in the FTIR spectra of the treated samples were detected.

Figure 3b shows that absorption bands of the hydroxyl group at 630 and 3573 cm^−1^ are present in all the spectra of the substituted samples.

The band of stretching vibration at 3573 cm^−1^ tends to broaden as the magnesium concentration increases, while the intensity of the libration band at 630 cm^−1^ decreases. This fact suggests that the concentration of hydroxyl groups in the OH channel upon substitution does not change, but the presence of the substituent in the nearest environment dampens the libration vibration of a hydroxyl group. A slight shift of absorption bands at high concentrations of the substituent is observed, but the magnitude of this shift is comparable to the measurement error, which makes it impossible to draw any conclusions.

To understand how the IR spectra should change when calcium cations are replaced with magnesium cations, the IR spectra of substituted and unsubstituted hydroxyapatite were simulated. The results are given in Section 3.3.

### 3.2. DFT Modeling

#### 3.2.1. Unit Cell Parameters

Figure 4 shows the results of calculations of the parameters and the volume of the hexagonal HAP unit cell, where Ca1 and Ca2 atoms were substituted with a different amount of x magnesium atoms after the relaxation of the structure to the equilibrium state. The introduction of magnesium into the crystal lattice of the HAP leads to a decrease in the parameters of the unit cell (Figure 4a), i.e., to the compression of the unit cell (Figure 4b), and to the non-equality of the cell parameters *a* and *b*. This causes the loss of the initially presented HAP hexagonal symmetry (*P*6_3_ group), where *a* = *b*. The most noticeable differences between the *a* and *b* parameters are observed at low concentrations (x < 0.5) of Mg/Ca2 substitutions (Figure 4a). In the case of low concentrations of Mg in the positions of Ca1, the *a* and *b* parameters are almost equal. Furthermore, the discrepancy between the parameters *a* and *b* for Mg substitutions in the Ca1 can be noticed at the concentrations x~2, while for Ca2 substitutions, these parameters are almost equal in this concentration range.

At a concentration x = 1.5, significant differences between *c* parameters for Mg/Ca1 and Mg/Ca2 can be noticed. This indicates significant changes in the positions of atoms in the unit cell. From a comparison of the calculation results and the obtained experimental data (Figure 4a), it can be seen that the behavior of all parameters of the HAP unit cell upon the substitution have the same dynamic, i.e., decreasing with an increase in the Mg concentration.

The slopes of the experimental curves for the samples before and after heat treatment are different. As noted above, adsorbed and lattice water completely evaporate from the particles after heating to 500 °C without any phase transformations. But some structural changes occur after heating, which lead to different dynamics of the changes in the lattice parameters. To assess the possible effect of the evaporation of hydroxyl groups from the OH channel, we calculated the unit cell parameters for the concentration of Mg x = 0, 0.5, 2 with the presence of one neutral OH vacancy V^(0)^_OH_ per supercell.

The obtained values are shown in Appendix A. The values of the cell parameters in this case turned out to be almost the same (taking into account the errors). But the presence of such a single OH vacancy significantly altered the electron band structure so that an additional energy level appears in the band gap (see Section 3.4).

Figure 4b shows that, for the calculated data, the values of the unit cell volume for the position of magnesium in the Ca1 and Ca2 state are almost the same for the concentration x ≤ 1. For concentrations x > 1, there are some deviations. From the experimental data, the closest values are observed for samples without heat treatment. After heat treatment, the linearity of the change in cell volume is broken at the point x = 1. Such changes cannot be caused by the formation of OH group vacancies upon heating, since, according to our calculations, the OH vacancy has very weak effect on the cell volume (Figure 4b).

The decrease in the unit cell parameters (Figure 4a) and volume (Figure 4b) is consistent with the difference between ionic radii of Ca^2+^ and Mg^2+^.

According to different sources, the ionic radii for Ca^2+^ and Mg^2+^ ions have the value r(Ca^2+^) = 0.99–1.14 Å, while for Mg^2+^, it is smaller: r(Mg^2+^) = 0.69–0.89 Å [39,70,71,72]. This difference is responsible for the observed shrinkage of the cell.

#### 3.2.2. Substitution in Ca1 Position

In the case of calcium substitution in the Ca1 position, the OH channel gradually narrows as the amount of magnesium increases and the distance from the *c*-axis of the OH channel to the ions in the Ca1 and Ca2 positions decreases (Figure 5). Characteristic changes in both OH channels upon substitution to 16 Mg/Ca1 are seen in Figure 5a,b.

##### Distances from OH Channel to Atoms in the Ca1 Positions

The changes that took place in the channel upon the substitution can be characterized by the distances between the atoms in the Ca1 positions and hydrogen atoms of OH groups (they lie in approximately the same plane with respect to the *c*-axis and have close *z* coordinates). Figure 5c,d show the diagram of changes in r(H–Mg/Ca1) distances in the supercell upon nMg/Ca1 substitution for different numbers of magnesium atoms *n* per supercell, where *n* = 4, 8, 12, 16.

The diagram shows that this substitution results in the contraction of the OH channel with some symmetry. In this case, the orientation of OH groups in the OH channel practically does not change. Detailed values of the r(H–Mg/Ca1) distances for a number of atoms lying in two *z* planes with the corresponding H atoms are given in Appendix A. In the initial state, the r(H–Ca1) distances are on the order of ~5.5 Å (Figure 5d). As the Mg concentration increases, the HAP-Mg structure gradually relaxes to the equilibrium state for each new value of the number *n* of substituted nMg/Ca1 and reaches the maximum contraction up to r(H–Mg/Ca1)~4.0 Å.

On average, a decrease in this distance reaches values on the order of ∆r = 0.25–0.35 Å for substitutions 12 Mg/Ca1 and 16 Mg/Ca1, which correspond to Mg concentration x = 1.5, 2.0.

As the number of substituted Mg ions successively increases, some asymmetry is observed in these distances along different axes lying in different planes of the OH channel cross-section. This is especially noticeable for Mg concentrations x = 1 (8 Mg/Ca1) (green line in the diagram in Figure 5d). When all positions of substitutions of Mg/Ca1 ions around the axis of the OH channel become occupied (the case of 16 Mg/Ca1, x = 2), the symmetry is practically restored (last column in Appendix A). The Mg ions in Ca1 positions at a Mg concentration x = 2 already strongly interact with the oxygen ions of the surrounding PO_4_ groups (Figure 5b), forming saturated ionic bonds up to an amount corresponding to their coordination number, which is equal to 6 [72].

##### Distances from OH Channel to Atoms in Ca2 Positions

Figure 5e,f show that the distances from the axis of the OH channel to the atoms in the Ca2 positions located in the nearest environment of the OH channel also change while the substitution of Mg/Ca in the Ca1 position was made. In this case, the distances were estimated in the cross-sections of other *z* planes, which pass through the oxygen atoms of the hydroxyl groups and lie in approximately one *z* plane with calcium or magnesium in positions with coordinates *z* = 12.00–12.27 Å (see Appendix A). Similarly, there are sections in the second plane Z with coordinates z = 8.58–8.28 Å, which cross calcium or magnesium atoms with oxygen atoms of OH groups. In this case, distances r(O–Ca2) from the OH channel axis to the positions of Ca2 atoms were obtained. All these data are given in Appendix A.

Some asymmetry was observed during the compression of Ca2 ions around the axis of the OH channel in different directions in the planes of the cross-section of the OH channel. When all positions of substitutions of Mg/Ca1 ions around the axis of the OH channel become occupied (the case of 16 Mg/Ca1 substitution, x = 2), then the symmetry improves slightly, and some reverse relaxation of Ca2 positions takes place (Figure 5f). Initial distances have average values of the order of r(O–Ca2) ~ 2.4 Å and decrease to values of r(O–Ca2) ~ 2.2 Å. On average, the total decrease in this distance r(O–Ca2) reaches values ∆r = 0.05–0.2 Å for 12 Mg/Ca1 substitution (x = 1.5), and then slightly expands back by ∆r = 0.02–0.05 Å for 16 Mg/Ca1 substitution (x = 2). These changes occur symmetrically around the OH channel, giving a uniform compression of the cell.

At high concentrations of Mg (x = 2 at *n* = 16 for 16 Mg/Ca1), the orientation of OH groups in the OH channel is partially distorted in some places. It should affect the librational vibrations of the OH group, which appear in the IR spectrum and were observed experimentally (Figure 3). Such deviations in the orientations of OH groups may affect the change in the behavior and cell parameters of the HAP-Mg crystal lattice, causing both the appearance of inequality of the *a* and *b* parameters and non-linearity in the cell volume change (Figure 4). However, since these minimum distances remain at the level of r > 2.25 Å (see Appendix A), no direct chemical bonds are formed in this case (Figure 5e). More precisely, according to the data from [72,73], the distance between the Ca^2+^ ion and the oxygen ion in oxides is d = 2.43 Å.

Using the value r(O^2−^) = 1.3 Å [72], we obtain r(Ca^2+^) = (2.43–1.3) Å = 1.13 Å. Taking into account that the coordination number of calcium is 8, in the Magnus–Goldschmidt model [72], we find that it will be r(Ca^2+^)~0.95 Å, and then d = r(O–Ca2) = (0.95 + 1.3) Å = 2.25 Å, i.e., *r_c_*~2.25 Å, is the critical value, where a chemical bond occurs. In our case, the minimum calculated distance turns out to be r(O–Ca2) = 2.30–2.35 Å (Appendix A), which is greater than *r_c_*. Thus, a direct ionic bond does not yet arise here.

##### Distances between Mg and O Ions from PO_4_ Groups

In this case, there are changes in the distances from the substituted Mg ions in the Ca1 positions to the surrounding oxygen ions O of the PO_4_ tetrahedra and the appearance of direct Mg–O chemical bonds (Figure 5b and Figure 6). An increase in Mg concentration in Ca1 positions causes a change in these distances r(Ca1/Mg–O) from Ca or Mg ions (with numbers #28, #30, #32) to the surrounding oxygen ions of PO_4_ groups. This leads to the achievement of critical distances *r_c_*, at which the formation of first three and then all six Mg–O ionic bonds (corresponding to the coordination number 6 for magnesium) is possible (Figure 6a–c). In this case, the chemical bond length r(Mg–O), determined by the ionic radii of oxygen (r(O^2−^) = 1.30–1.32 Å [72]) and magnesium (r(Mg^2+^) = 0.69–0.89 Å [39,68,69,70,71,72] with a coordination number of 6) can reach r(Mg–O) = *r_c_* ~ (2.09–2.19) Å, when their stable chemical ionic bond is formed. This leads to the formation of all six Mg–O ionic bonds upon the substitution of 16 Mg/Ca1 (x = 2).

Figure 6c,d show distances around the Mg #30 ion. Diagrams of similar changes in distances around Mg #28 and Mg #32 are shown in Appendix A. Changes in r(Mg–O) distances for different amounts of Mg are given in Appendix A.

We also note that the relative changes (∆r/r_0_) in the distances between ions during the cell compression reach ~15% here (Appendix A). These results are consistent with various data from Ren [39], Makshakova [56], and other authors [55,57,58]. However, in our work, the data obtained are much more complete and detailed, since a wider range of substituent concentration is covered here, more detailed calculations were carried out, and the models and DFT methods used are more accurate and correct, which allows us to more adequately assess the changes that occur during the substitutions under study.

#### 3.2.3. Substitution in Ca2 Position

##### Distances from OH Channel to Atoms in Ca2 Positions

Substitutions of Mg/Ca in the Ca2 positions lead to slightly different changes in the behavior of the characteristic distances in the HAP-Mg structure of its supercell and unit cell. First of all, this is due to the fact that the Ca atoms/ions in the Ca2 positions are noticeably closer to the OH channel axis, and because the cell parameters decrease and the entire volume is compressed with an increase in the substituted Mg/Ca2 atoms/ions, they quickly move away from the OH channel axis at distances at which O^2−^ oxygen ions from the OH group are able to form a stable chemical bond with the magnesium ion.

As noted above, the ionic radii of oxygen determine the chemical bond lengths r(Ca–O) and r(Mg–O), which here vary with different amounts of Mg ions replacing Ca at the Ca2 position. Magnesium has an ionic radius r(Mg^2+^) = 0.69–0.89 Å, and its coordination number is 6 [70,71,72]. When the bond length between O and Mg decreases to the values r(Mg–O) = 2.09–2.19 Å~*r_c_*, their stable chemical ionic bond is formed. The results of such nMg/Ca2 substitutions are shown in Figure 7 and Table 2 and Appendix A.

A noticeable asymmetric capture of O^2−^ ions by the nearest substituted magnesium ion Mg^2+^ located in the Ca2 position occurs, and this oxygen ion (and the entire corresponding OH group) is shifted to the side, which violates the symmetry and the entire one-dimensional structure of the OH channel. These distortions of symmetry make a significant contribution to the behavior of the cell parameters, especially in the range of magnesium concentrations x = 1.5 (*n* = 12) in the case of Ca2 substitution. Direct bonds of Mg ions with oxygen ions of the OH group arise here, which leads to their displacement from the axis of the OH channel to the side, and this immediately strongly violates the entire symmetry of the HAP-Mg structure. Appendix A shows a typical example of such changes.

In the upper part of Table 2, the distances in the left OH channel from one atom (#66), which is located in the Ca2 position near the axis of the OH channel, to all nearest O atoms (PO_4_ and OH groups) are given. These data show how the distances noticeably change as the magnesium concentration increases and the distances reach their critical minimum values *r_c_* = 2.09–2.19 Å, when a stable chemical (ionic) bond is formed. The number of direct chemical bonds Mg–O increases with increasing Mg concentration x.

The lower part of Table 2 shows the data on the distances (including their relative changes) in the right OH channel between atoms in positions Ca2 and oxygen ions from the corresponding OH group. It is seen that the presence of the Mg ion in the Ca2 position sharply reduces the distance r(O–Mg/Ca2). This leads to a shift of the entire OH group towards the position of the Mg ion, which causes the appearance of a noticeable asymmetry.

Moreover, the complete divergence of the two OH groups relative to each other here reaches distances on the order of ~0.64 Å (or ~0.32 Å relative to the axis of the OH channel) for the case of a Mg concentration x = 2 (16 Mg/Ca2).

Thus, in general, there is a significant reduction in the distances r(Mg–O) for the substituted Mg ions in Ca2 positions, and a direct ionic bond of Mg with oxygen from the OH group is formed, which occurs at *r* < *r_c_* ~ 2.09 Å. The shortening of distances r(Mg–O) in the OH channel reaches ∆r = 0.22–0.25 Å for the case of Mg concentration x = 2 at 16 Mg/Ca2 substitution (*n* = 16), and their total value reaches ∆r = 0.45–0.5 Å. All this, of course, affects the vibrations of bonds, which are reflected in the IR spectrum.

The characteristic relative changes in all these distances reach values on the order of ~10%, which also agrees in principle with some data from a number of works such as those by Ren [39] and other authors [55,56,57,58]. The width of the OH channel itself changes little.

### 3.3. Modeling of IR Spectrum

IR spectra were modeled for the case of single-unit cell atomic models of unsubstituted HAP (Figure 8a,b) and Mg-substituted HAP in which one out of four Ca1 atoms (Figure 8c,d) or one out of six Ca2 atoms (Figure 8e,f) was substituted with Mg. This variant of substitution is equivalent to the case of *x* = 1. The IR spectra were calculated according to the method described in our previous work [34]. The modeled IR spectra for Mg-substituted HAP are shown in Figure 9. To identify peaks in the modeled IR spectrum, these peaks were compared with the vibrations, taking into account the calculated atomic displacements. The positions of bands in the modeled spectra in comparison with the experimentally observed ones are shown in Table 3.

The width of the bands is not reproduced, since the spreading constant of 1 meV is used for calculations, and the systematic shift between the theoretical and experimental positions of these bands has already been mentioned by other authors [74,75].

According to the results presented in Figure 9 and Table 3, the shift in the frequency of the OH group librational vibration mode (ν_L_) occurred when calcium atoms were substituted in columns far from the OH channel (the case of Ca1 substitution). A weak additional band of librational vibrations at 700 cm^−1^ also appears. The probability of the experimental registration of this band is very low. In addition, a shift of the ν_1_ vibration of PO_4_ tetrahedron to the low frequency is observed.

When the calcium atom is replaced in the OH channel wall (the case of Ca2 substitution), two stretching (ν_S_) and three librational (ν_L_) modes of the OH group are observed in the modeled spectrum due to the nonequivalence of OH groups position located in the calcium channel (Figure 8). Moreover, the wavenumber for vibrations of the OH group located closer to the Mg atom, i.e., more strongly shifted from the center of the OH channel, turns out to be higher than for other OH groups. The librational vibrations of shifted OH groups also have higher wavenumbers than the unshifted groups. As for the phosphate group, in the case of the Ca2 substitution, there is a positive shift of one of the stretching bands ν_3_.

This corresponds to the most characteristic changes in the distances r(Mg–O) after the replacement of calcium by magnesium Mg/Ca in the Ca2 position inside the OH channel (i.e., near the walls of the OH channel). These distances are changed from about 2.44Å to 2.09 Å (see above Section 3.2.3 and Table 2). In this case, the bound pairs of Mg-O ions are shifted relative to the axis of the OH channel by ~0.30 Å, as noted in the previous section.

Thus, it can be concluded that the introduction of Mg into the Ca2 position gives a more noticeable effect on the IR spectrum, which manifests itself both as a shift in the absorption bands and in the appearance of additional bands belonging to the OH group.

Comparison of the simulated IR spectra (Figure 9) with the experimentally observed ones (Figure 3) suggests that, in the experiment, Ca2 substitution is predominantly observed. First of all, this is indicated by the behavior of the OH group’s absorption bands. A significant decrease in the intensity of the librational mode at 630 cm^−1^ is accompanied by its widening (Figure 3), which can be explained by duplicating the band. The same behavior is observed in the simulated IR spectrum (Figure 9). A slight shift and appearance of a shoulder for the stretching mode of the OH group at 3573 cm^−1^ in the experiment (Figure 6) also agrees with the simulated spectrum (Figure 9). The shift of the absorption band ν_3_ of the PO_4_ group from 1088 cm^−1^ to 1092 cm^−1^ is consistent with the shift in the simulated spectrum (Table 3). A slight shift and widening of the ν_1_ mode of the P–O bond can be connected with Ca1 substitution (Table 3).

### 3.4. Changes in the Electron Band Structure

The spatial changes in the structure of HAP-Mg unit cells, which arise upon substitutions of Mg/Ca, also lead to a change in the electronic band structure, in particular, in the band gap E_g_.

The information about E_g_ is critical for understanding the semiconductor and optoelectronic properties of modified HAP-Mg. Figure 10 shows the calculated dependencies of the band gap E_g_ on the number of Mg/Ca substitutions in the HAP structure. The red and blue curves in Figure 10 were obtained using the PBE exchange-correlation functional, which, like other local and semi-local functionals, underestimates the values of E_g_ [76]. The red curve was obtained without the relaxation of the parameters of the cell containing the defect. This calculation is justified in the case of a low defect concentration, when a cell with a defect is located among a large number of defect-free cells and the presence of defects does not affect the lattice parameters. However, since the X-ray diffraction analysis reveals changes in the HAP-Mg lattice parameters, the E_g_ values were calculated for fully relaxed structures (with the relaxation of atomic positions and cell parameters, which, as described in Section 3.2.1, reduces these parameters and compresses the cell volume).

The band gap calculated for fully relaxed cells using the PBE functional weakly depends on the impurity concentration and is ~5.6 eV, which is ~0.4 eV higher than the band gap of defect-free HAP. A more accurate estimate of the band gap width using the HSE functional (the black curve in Figure 10) gives qualitatively the same behavior: the gap width weakly depends on the defect concentration and is ~0.5 eV higher, compared to E_g_~7.1 eV [26] in defect-free HAP. The E_g_ values obtained with the HSE functional, as expected, are ~1.7 eV higher than those obtained with the PBE functional [24,25,26,31]. The trends in the changes in E_g_ with an increase in the number *n* of nMg/Ca substitutions are different depending on the positions of the Ca atoms in which they occur: substitutions in Ca1 positions tend to increase E_g_, while substitutions in Ca2 positions noticeably decrease E_g_. This effect was discovered for the first time and has not yet been experimentally investigated.

The Mg/Ca substitution does not lead to the appearance of additional impurity energy levels in the band gap, but such levels can arise due to the presence of some other defects [24,25]. In particular, we considered a combination of Mg/Ca substitutional defects and OH vacancies for the case of *n* = 0, *n* = 4, and *n* = 16 nMg/Ca. In the case of an OH vacancy (OH-vac), a partially populated impurity energy level arises, for which electron transitions are possible both from the valence band to it and from it to the conduction band [24,25,26] (see Figure 11). Here, E_i_ is the energy of the additional impurity level of a neutral OH-vac, or V^(0)^_OH_ [26], which defines the electron transitions as E_1_ = E_i_ − E_V_ (where E_V_ is the top of valence band, E_V_ = E_HOMO is the highest occupied molecular orbitals, and E_2_ = E_C_ − E_LUMO—the lowest unoccupied molecular orbital). This corresponds to the appearance of an E_i_ level inside the band gap, in this case equal to E_i_ = E_F_ to the Fermi level, since, according to our DFT calculations for this case, it is populated by ½ (Figure 10 and Figure 11).

The values obtained for the fundamental forbidden gap E_g_* (after the insertion of the additional energy level E_i_ induced by OH-vac) and the energy distances to the additional impurity level from the top of the valence band E_1_ are marked in Figure 10b by dots. The results of DFT calculations show that the width of the fundamental gap E_g_* here can be increased by ∆E_g_* ~0.4 eV through the formation of the OH vacancy in unsubstituted HAP (x = 0). Then, in the presence of Mg/Ca substitutions, this value gradually decreases to ∆E_g_*~0.2 eV for Mg/Ca substitutions at the concentration x = 2, for substitutions in both the Ca1 and Ca2 positions. In this case, the position of the impurity level E_1_ in the forbidden gap varies from ∆E_1_~0.4 eV to ∆E_1_~1.4 eV depending on the number and Ca positions of substitutions.

Note that the values of the characteristic energies of photoelectron transitions E_1_ obtained here for HAP-Mg turn out to be close to the results of calculations of defects such as OH-vac in HAP [24,25,26] and lead to similar changes in its photoelectronic properties compared to pure HAP.

Appendix A provides more detailed data on all the main calculated values of these energies, E_g_, E_g_*, E_1_ and E_2_, as well as their relative changes at different concentrations x of magnesium Mg.

Thus, magnesium substitutions for calcium atoms in positions Ca1 and Ca2 lead to an increase in the band gap by ~0.4 eV compared to defect-free HAP. In this case, depending on the positions of calcium, as the number of substitutions increases, weak tendencies of an increase (for Ca1) and a decrease (for Ca2) in the band gap are noticeable. A combination of substitutions and intrinsic defects can reverse this trend. In particular, in the presence of an OH vacancy, the relative band gap changes insignificantly by ∆E_g_*~0.1 eV, while the position of the impurity level in the band gap turns out to be dependent on the number and positions of Mg/Ca substitutions.

### 3.5. Change in Bulk Modulus B

Based on the optimized HAP and HAP-Mg structures, DFT calculations were carried out in this work, according to known methods of similar calculations [12,24,25,26,27]. In our case, we obtained the value B = 86.606 GPa for the initially pure, defect-free stoichiometric HAP (P6_3_). The known experimental data are B = 82.6–89.0 GPa [77,78], and the data from a series of various DFT calculations are B ~ 82 GPa [24], 86 GPa [26], 88 GPa [34]. After Mg/Ca2 replacement of Mg concentration x = 2 (*n* = 16 for the case of 16 Mg/Ca2), the modulus value decreased up to B = 80.62 GPa. This indicates a decrease in the mechanical strength of HAP-Mg at such Mg concentrations.

On the other hand, when substituting Mg in Ca1 position at the same Mg concentration x = 2 (for *n* = 2 of 16 Mg/Ca1), similar DFT calculations performed in this paper give an increased value for B = 87.55 GPa (with the derivative value B’ = 5.17). This indicates an improvement in the mechanical strength of HAP-Mg in this case.

### 3.6. Formation Energy of Mg/Ca Substitutions in Different Ca1 and Ca2 Positions

The calculation of the formation energy E_f_ of nMg/Ca substitutions for different values *n* of the amount of Mg, and in different positions of Ca1 and Ca2, was estimated according to the formula
E_f_ = E_tot_ − E_HAP_ − *n*·[μ(Mg) − μ(Ca)](2)
where E_HAP_ is the total energy of the net initial HAP, taken for a 2 × 2 × 2 = 8 supercell; E_tot_ is the total energy of HAP-Mg, calculated after the relaxation of the supercell with a given number of substitutions *n* of Ca atoms for Mg atoms (for various chosen positions of Ca atoms, nMg/Ca1 and nMg/Ca2); and μ(Mg) and μ(Ca) are chemical potentials of Mg and Ca ions calculated for standard phases of magnesium and calcium, respectively. Their values are as follows: μ(Mg) = −1478.7338 eV and μ(Ca) = −1003.7562 eV. Accordingly, their difference is as follows: [μ(Mg) − μ(Ca)] = −474.9776 eV.

In the case of an OH-vac, Equation (2) is replaced by
E_f_ = E_tot_ − E_HAP_ − *n*·[μ(Mg) − μ(Ca)] − [μ(O) + μ(H)]

If we are interested in the case of the energy of substitution formation in an HAP crystal that already contains a vacancy (e.g., one neutral OH vacancy), then instead of E_HAP_ − [μ(O) + μ(H)], the energy of the crystal E_HAP-OH_ should be used. In our case, the DFT calculations performed give the following values: E_HAP-OH_ = −179,629.65617 eV, E_HAP_ = −180,083.63773 eV, and, accordingly, E_HAP_ − E_HAP-OH_ = −453.98156 eV.

For analysis and comparison with various known data, the formation energies E_f_ are reduced to one formula unit (f.u.) HAP-Mg. The 2 × 2 × 2 supercell model used contains eight HAP unit cells, each of which includes two formula units. Therefore, the energy values finally presented are divided by the number of all formula units, equal to 8 × 2 = 16. Figure 12 and Table 4 show the calculated behavior of E_f_ as a function of Mg concentration at different Ca1 and Ca2 substitution positions.

This dependence demonstrates an ambiguous change in E_f_ at different substitution positions: in the region of low Mg concentrations, the values E_f_(Ca1) > E_f_(Ca2).

For the concentration of Mg x = 0.5, there is a significant minimum of E_f_(Mg/Ca2)~0.096 eV/f.u. In this case, the average value E_f_(Mg/Ca1)~0.105 eV/f.u., which varies from ~0.100 to 0.115 eV/f.u.

It is noteworthy that, at a concentration x = 1 (*n* = 8 for nMg/Ca1 and nMg/Ca2 substitutions), a certain critical point is observed, with the minimum for values of E_f_(Mg/Ca1) ~0.1 eV/f.u. and an inflection in the behavior of E_f_(Mg/Ca2) ~0.1025 eV/f.u. With the increase in concentration, the E_f_(Mg/Ca2) values slightly decrease and remain at approximately the same level at ~0.102 eV/f.u., while the E_f_(Mg/Ca1) values gradually increase and reach the maximal value of E_f_(Mg/Ca1) ~ 0.12 eV/f.u. at a Mg concentration x = 2 (*n* = 16).

The presence of OH-vac did not actually affect the substitution in the Ca2 position (it only slightly increases energy but within the error bar), whereas, upon substitution in the Ca1 position, the presence of the OH vacancy reduced the formation energy E_f_ (by almost ~0.005 eV/f.u.) at both the chosen magnesium concentrations x = 0.5 and x = 2.

Using the distribution of an ensemble of atoms (ions) N~N_0_·exp(−E_f_/kT), where the energy kT = ~0.0257 eV (the values of kT at 298 K are room temperature), we can estimate how strongly the differences in the energies E_f_ upon Mg/Ca substitutions in the Ca1 and Ca2 positions affect the probability of their formation. In this case, our estimates showed that the probability of Mg occupying the Ca2 position is about two times higher than the Ca1 position. But this does not exclude the statistical probability of Mg occurrence in the Ca1 position. This only means that this probability is two times less.

In all these cases, it should also be taken into account that the error resulting from these DFT calculations is estimated by us at the level of ~0.014 eV/f.u.~0.5·kT.

### 3.7. Discussion on the Formation Energies of Mg/Ca Substitutions

Thus, in this case, the energy of formation E_f_ for substitutions in the Ca1 and Ca2 positions differs not so significantly, and both types of defects can coexist in a real crystal. According to our estimates, the possible number of substitutions in the Ca2 positions is approximately twice as large as in the Ca1 positions.

It should also be noted that the order of magnitude of the formation energy of these substitutions obtained in our work is E_f_ ~ 0.1…0.12 eV/f.u., which agrees with the data from other authors. For example, Fetodkin has E_f_ ~ 0.16 eV/f.u. [54]. But the data available here from different authors are at the same time rather contradictory.

For example, in Laurencin [57], the energy difference between substitutions in different positions turned out to be on average only ~0.25 eV in favor of Mg/Ca2, which confirms the higher stability of the structure of HAP with Mg in media where substitution at the Ca2 position exists in a larger composition range, while in Ren [39], in their supercell model for a solid solution with x = 0.5 Mg, the energy difference between Mg/Ca1 and Mg/Ca2 was 0.012 eV in favor of Mg/Ca1 substitution. For x = 1 Mg substitution, the energy difference here was ~0.077 eV, which is even more favorable for Mg/Ca1 substitution. That is, the results of [39] show that substitution in the Ca1 position is energetically more favorable here. But the results of Mashkakova’s work [56] give a difference in the energies of Mg formation in the Ca1 position 0.042 eV higher than for Mg in Ca2. That is, although not much here, it turns out to be more profitable to replace Mg in the Ca2 position. Note also that Saito [58] also gives different data for different types of positions of substituted ions. Thus, there are rather contradictory data in the literature on the values of the energy of formation of Mg/Ca substitutions in HAP-Mg.

However, these contradictions can be removed if we consider a wider range of possible substitutions in terms of the concentrations of substituted ions (in particular, nMg/Ca1 and nMg/Ca2) and take into account their statistical differences, as is performed in this our work. Then, it becomes clear that under different conditions and concentrations of nMg/Ca in different Ca1 and Ca2 positions, the formation energies E_f_ can indeed differ.

According to our calculations, in the range of concentrations x = 1 Mg, E_f_ can almost coincide, and at concentrations x = 0.5 Mg and x = 2 Mg (and more), Mg substitutions in the Ca2 positions have some advantage. But at the same time, the entry of the Mg ion into the Ca1 position is not statistically absent, and it is also possible, although with a twice lower probability.

Thus, both variants of substitutions and their coexistence are possible here, depending on the specific conditions of the experiment. It is important to note that statistically random formation of defects, such as OH-vac, can also have an effect, which, in principle, can also occur at annealing temperatures that do not exceed 500 °C. The results of our calculations in this work show that such an effect can be more significant in the case of substitutions in the Ca1 position (see Figure 12 and Table 4). In this case, the formation energy E_f_ decreases by ~0.005 eV/f.u., which increases the probability of Mg substitution in the Ca1 position.

## 4. Conclusions

The high-precision hybrid DFT calculations performed in combination with experimental studies in a wide range of Mg/Ca substitutions in the HAP lattice using a 2 × 2 × 2 supercell model showed that the parameters and volumes of the HAP unit cells gradually decrease with the increase in the number of substitutions, which is consistent with the data from other authors. A decrease in the lattice parameters by ~0.04 Å and the unit cell volume by 6–8 Å^3^ was observed upon substitution with one magnesium ion. A more detailed consideration revealed the dependence of the parameters and volumes of the cells from the substitution position (Ca1 or Ca2), which are especially significant in the Mg concentration range of x = 1–1.5.

It was found that formation energies of substitutions have non-monotonic behavior, which depends on the position of the replaced Ca atom: Ca1 or Ca2. The advantages of Mg/Ca substitutions in different Ca1 and Ca2 positions turn out to be different and depend significantly on the concentration x of introduced Mg. This complex dependence and the possibility of the coexistence of the substitutions of both types may be the reason for the ambiguity and inconsistency in the values of the formation energy available in the literature. We show that, on average, the Ca2 position has only a two times higher probability of Mg/Ca substitutions than the Ca1 position with a formation energy advantage of ~5 eV/f.u/substitution at a Mg concentration x = 0.5, while at a Mg concentration x = 1, the formation energies are almost equal for both types of substitutions, and Mg substitution can coexist here at the positions Ca1 and Ca2.

The possible presence of OH vacancies lowers the formation energy of Mg/Ca substitution in the case of Ca1 replacement by 5–8 meV/f.u./substitution. We also demonstrate that OH vacancies significantly change the electronic properties of HAP: the interstitial half-occupied energy level appears in the band gap, which alters the photoelectronic properties of the material. In the absence of OH vacancies, the band gap changes depending on the position of substitution (Ca1 or Ca2). At the lowest considered Mg concentration x = 0.125, the band gap increases by ~ 0.4 eV for both Ca1- and Ca2-substituted HAP, compared to that of bulk HAP. With the increase in the concentration, the band gap non-monotonically increases (by ~0.2 eV) for Ca1 substitutions but monotonically decreases (also by ~0.2 eV) for Ca2 substitutions. However, the position of the interstitial level is quite sensitive to the Mg doping, so the electron transition energies may differ in a wide range, from ~1.5 to ~7.5 eV, which points to the possibility of controlling the optical properties of the material.

The mechanical properties of Mg/Ca-substituted HAP also depend on the Mg position. We show a decrease in the bulk modulus upon substitutions in the Ca2 position and an increase upon Mg substitution in the Ca1 position.

It is shown that the distortion of the HAP unit cell symmetry under Mg/Ca substitution is explained by the changes in the distances between the Ca–O, Ca–H, and Mg–O ions. These data are consistent with measurements and DFT calculations of IR spectra showing changes in the frequencies of bond vibrations of HAP-Mg structures. A more pronounced change in the spectrum was detected for Ca2 substitution. This is due to a distortion of the OH groups’ position and the shift of the formed Mg–O pair relative to the axis of the OH channel. These changes are clearly visible in the experimentally obtained IR spectra, as well as in the simulated one.

In general, the results of this work represent new data that are of significant importance for the development of the technology for the production of HAP with Mg/Ca substitution, which are important for biomedical applications. The obtained information about the structural and energetic properties are important for the understanding of the interaction mechanisms of the HAP-Mg material with living bone tissue when it is used as a filler for bone defects or for implant coating.

## Figures and Tables

**Figure 1 materials-16-05945-f001:**
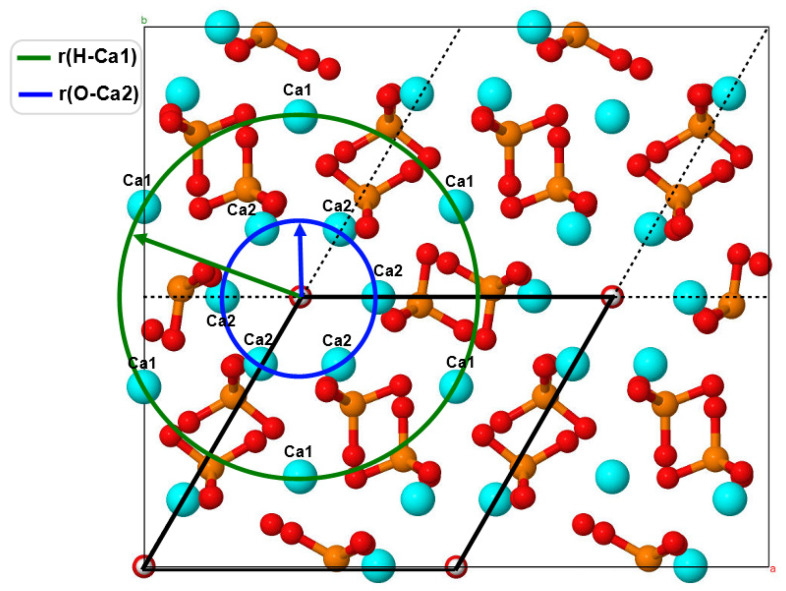
The orthorhombic HAP supercell model (352 atoms) and its geometric relation to the model of a hexagonal unit cell (44 atoms, marked by black thick lines; the black dashed line shows this unit cell periodically repeating) viewed from the Z-axis. The two types of calcium positions, Ca1 (green circle) and Ca2 (blue circle line), are located around the axis of the OH channel. The corresponding circle radii are r(H–Ca1) and r(O–Ca2). Notation: light blue—Ca, green—Mg, red—O, brown—P, white—H atoms.

**Figure 2 materials-16-05945-f002:**
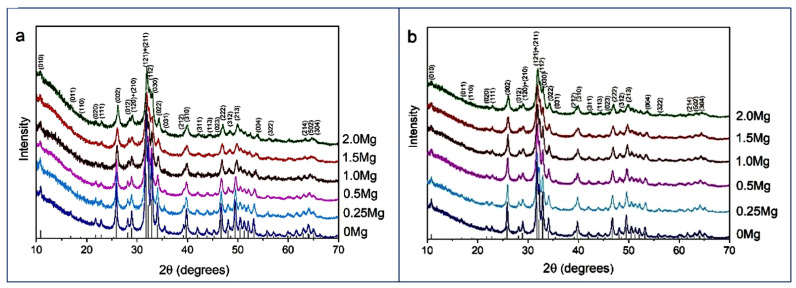
XRD patterns of synthesized powders with different amounts of introduced magnesium before (**a**) and after (**b**) heat treatment at 500 °C. The bar graph corresponds to the position of the HAP reflections from card PDF 01-76-0694.

**Figure 3 materials-16-05945-f003:**
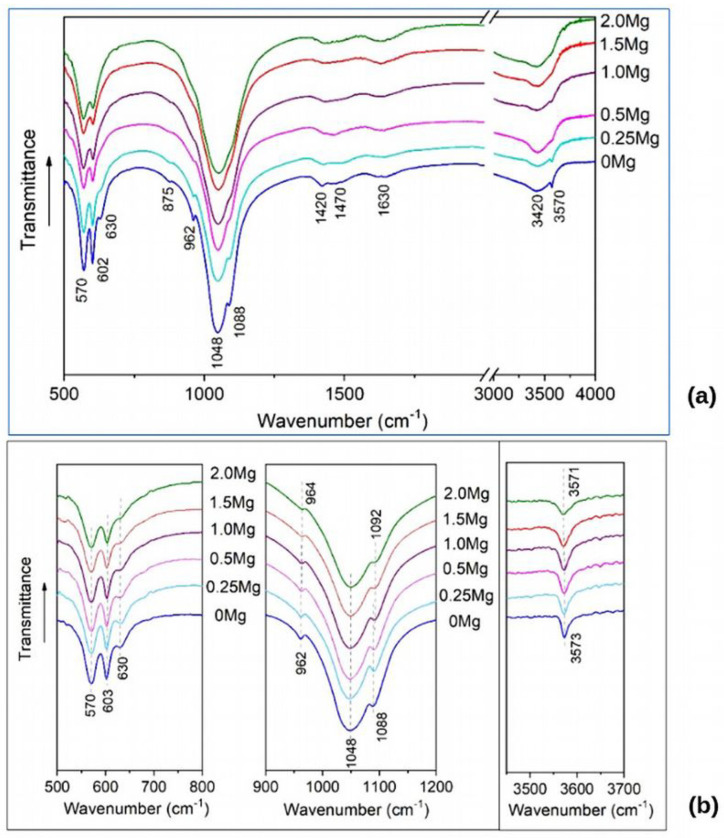
FTIR spectra of the samples synthesized with the addition of different amounts of magnesium: (**a**) as synthesized; (**b**) after heat treatment at 500 °C.

**Figure 4 materials-16-05945-f004:**
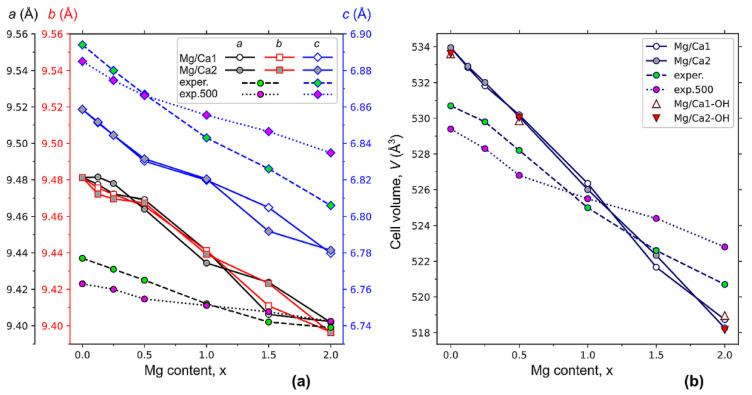
Changes in the parameters (**a**) and volumes (**b**) of the hexagonal cell of HAP at different concentrations x of Mg/Ca substitutions in Ca1 and Ca2 positions in comparison with the experimental data for the synthesized samples (exper.) and after heat treatment at 500 °C (exp.500). The left (**a**) shows the unit cell parameters of HAP in the P6_3_ hexagonal phase: a, b, and c in Angstrom units.

**Figure 5 materials-16-05945-f005:**
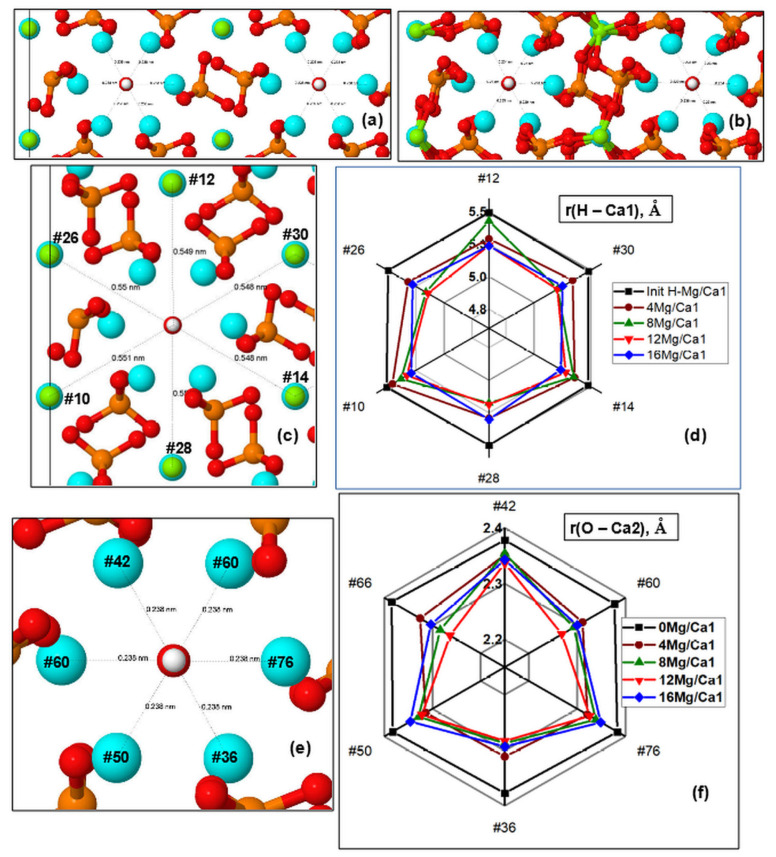
HAP supercell with 16 Mg atoms in the Ca1 positions in the *z* plane projection before (**a**) and after (**b**) relaxation, and diagrams of distances from the *c*-axis of the OH channel to marked atomic positions: (**c**) in Ca1 positions from H atoms of the OH group; (**d**) with different numbers of Mg atoms in Ca1 positions (nMg/Ca1, where *n* = 0, 4, 8, 12, 16); (**e**) in Ca2 positions from the O atom of the OH group; (**f**) with different numbers of Mg atoms in Ca1 positions (nMg/Ca1, where *n* = 0, 4, 8, 12, 16) in the supercell. All the distances on the diagrams are in Angstrom. Notation of atoms color (denoted themselves as circles) are the same as on Figure 1; the indicated numbers indicate the atom’s number in the HAP supercell of 362 atoms.

**Figure 6 materials-16-05945-f006:**
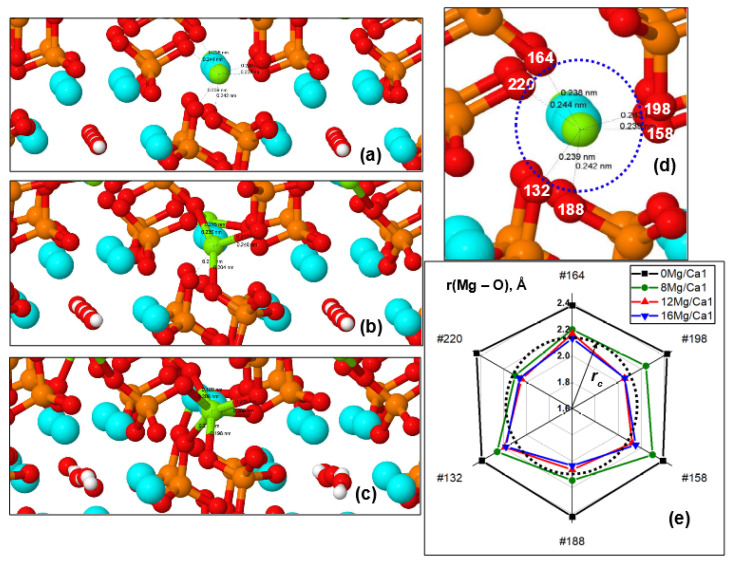
Change in distances and formation of Mg–O chemical bonds between Mg atom #30 in the Ca1 position and O ions of the nearest PO_4_ groups: (**a**) initial state at *n* = 8 (nMg/Ca1); (**b**) partial relaxation at *n* = 12 (nMg/Ca1) and formation of 3 Mg–O bonds; (**c**) complete relaxation at *n* = 16 (nMg/Ca1) and formation of 6 Mg–O bonds; (**d**) initial configuration and numbers of oxygen ions involved in the formation of bonds; (**e**) diagram of changes in distances r(Mg–O) around atom #30. Notation of atoms color (denoted themselves as circles) are the same as on Figure 1; the indicated numbers indicate the atom’s number in the HAP supercell of 362 atoms.

**Figure 7 materials-16-05945-f007:**
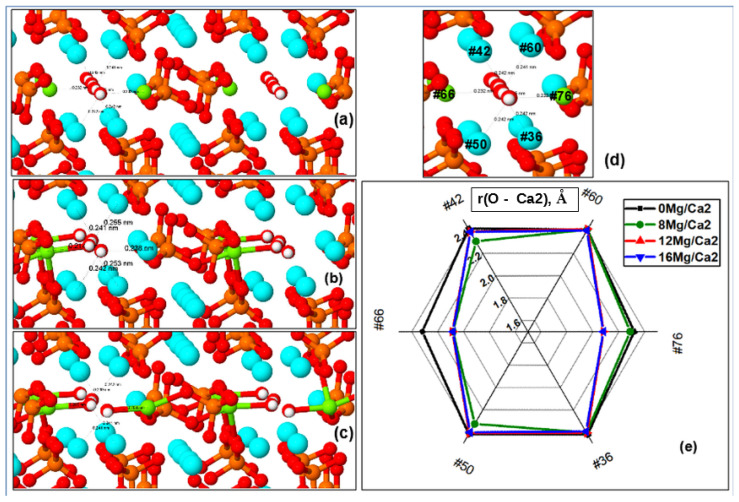
Change in the distances r(O–Ca2) between Ca and Mg atoms in the Ca2 position and O atoms of the OH group with an increase in the number of Mg atoms upon the substitution nMg/Ca2: (**a**) initial state for the case of 8 Mg/Ca2; (**b**) asymmetric displacement of O oxygen atoms in the OH channel and all OH groups for the case of 8 Mg/Ca2 with the formation of direct Mg–O ionic bonds (green); (**c**) symmetry distortion, when there is also a displacement of the oxygen ion O from the OH group to the other side for the case of 12 Mg/Ca2; (**d**) image of the left OH channel with the indication of numbers of atoms surrounding this channel in the initial state; (**e**) diagram of distances r(Mg/Ca2–O) for *n* = 8, 12, 16 after the relaxation in comparison with the initial state. Notation of atoms color (denoted themselves as circles) are the same as on Figure 1; the indicated numbers indicate the atom’s number in the HAP supercell of 362 atoms.

**Figure 8 materials-16-05945-f008:**
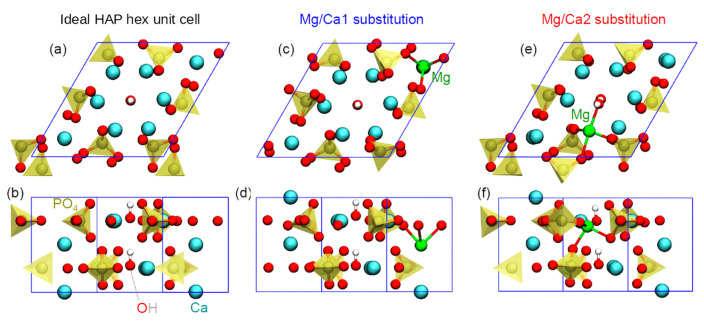
Model of unsubstituted hexagonal HAP unit cell (**a**,**b**) and Mg-substituted HAP unit cell with the replacement of one calcium atom in positions Ca1 (**c**,**d**) and Ca2 (**e**,**f**). Panels (**a**,**c**,**e**) show projection along the *c* axis, and panels (**b**,**d**,**f**) show projection along the *b* axis.

**Figure 9 materials-16-05945-f009:**
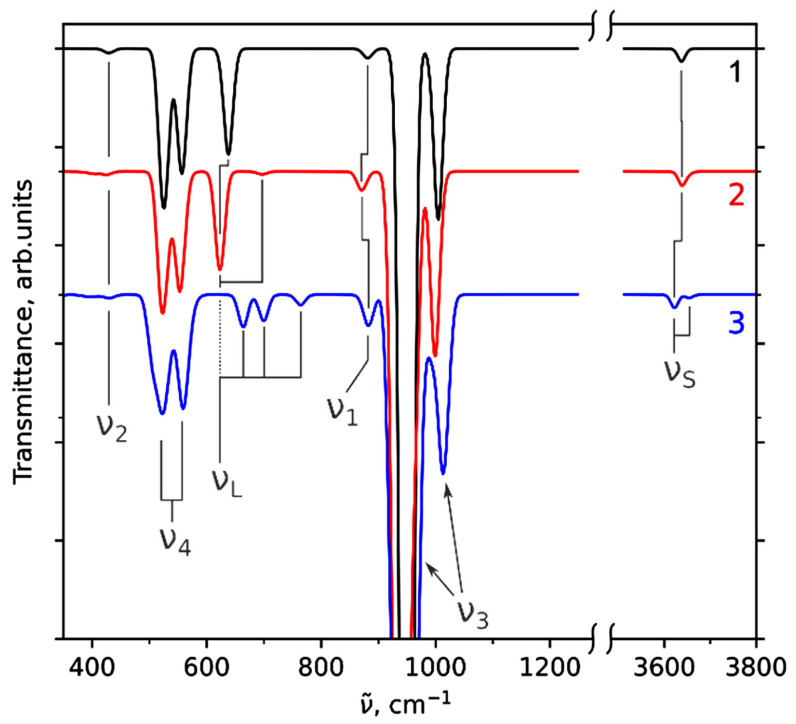
The simulated IR spectra of unsubstituted HAP (curve 1) and substituted HAP with the replacement of one calcium atom in positions Ca1 (curve 2) and Ca2 (curve 3) in accordance with Figure 8.

**Figure 10 materials-16-05945-f010:**
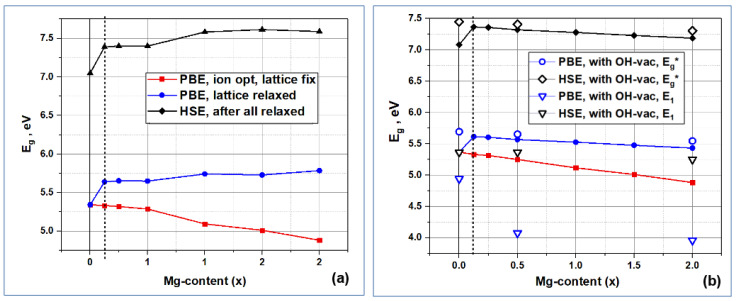
Change in the band gap E_g_ upon substitutions nMg/Ca in different positions of the calcium atom Ca: (**a**) in Ca1 positions; (**b**) in Ca2 positions. The effect of OH vacancy (OH-vac) on the E_g_ → E_g_* change is shown with points at selected concentrations x. We also show the energy E_1_ = E_i_ − E_V_ (E_1_ = E_i_ − E_HOMO), which is the energy distance from valence band top to the defect energy level in the band gap.

**Figure 11 materials-16-05945-f011:**
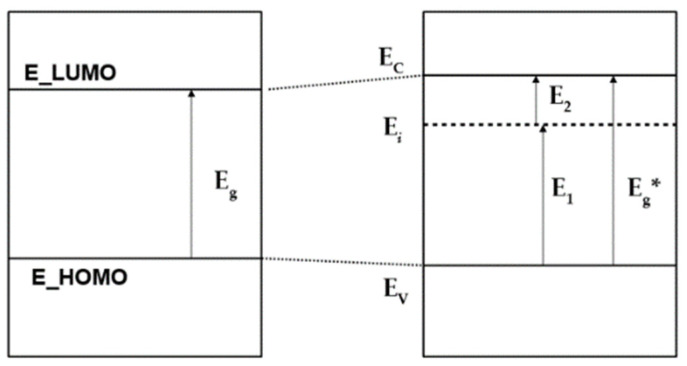
Change in the electron band structure of HAP-Mg in the presence of an OH-vac.

**Figure 12 materials-16-05945-f012:**
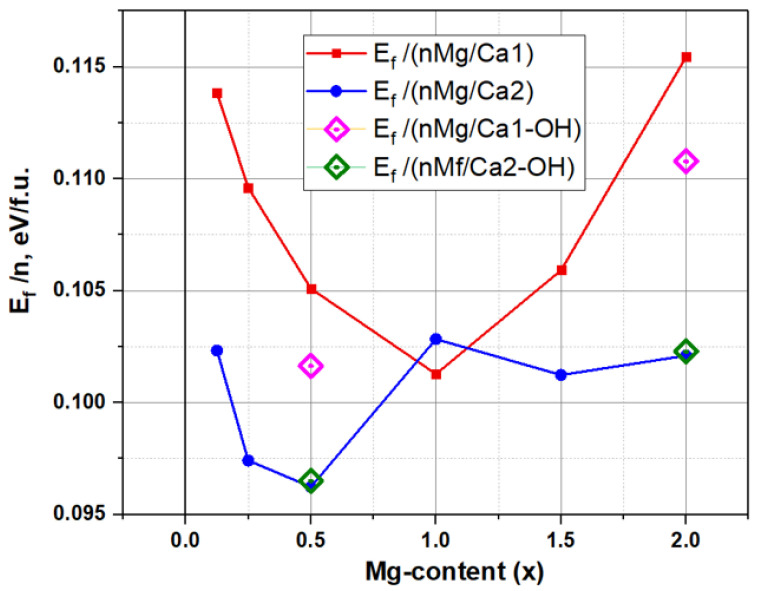
Dependence of the formation energy of Mg/Ca substitutions on the number of replaced magnesium ions *n* (nMg/Ca) in different Ca1 and Ca2 positions. The effect of OH-vac is marked with dots.

**Table 1 materials-16-05945-t001:** Structural characteristics of the HAP phase in the synthesized samples with different amounts of introduced magnesium (x) before and after heat treatment at 500 °C.

Sample Name	Mg Content, x	*a* (Å)	*c* (Å)	V (Å^3^)	Crystallite Size (nm)
Synthesized powders					
0 Mg	0	9.437 (1)	6.894 (1)	530.7 (2)	24.3 (2)
0.25 Mg	0.25	9.431 (1)	6.880 (2)	529.8 (2)	21.8 (2)
0.5 Mg	0.5	9.425 (2)	6.867 (2)	528.2 (2)	19.8 (2)
1.0 Mg	1.0	9.412 (2)	6.843 (2)	525.0 (2)	18.2 (4)
1.5 Mg	1.5	9.402 (2)	6.826 (3)	522.6 (3)	13.7 (4)
2.0 Mg	2.0	9.399 (2)	6.806 (3)	520.7 (4)	15.5 (4)
After heat treatment at 500 °C					
0 Mg	0	9.4230 (1)	6.8850 (1)	529.40 (2)	29.2 (2)
0.25 Mg	0.25	9.4201 (2)	6.8746 (4)	528.31 (5)	22.6 (2)
0.5 Mg	0.5	9.4146 (8)	6.8663 (6)	527.05 (9)	20.5 (2)
1.0 Mg	1.0	9.4111 (4)	6.8555 (5)	525.84 (6)	18.9 (2)
1.5 Mg	1.5	9.4076 (5)	6.8465 (5)	524.75 (7)	16.7 (2)
2.0 Mg	2.0	9.4024 (6)	6.8348 (6)	523.28 (8)	16.4 (2)

**Table 2 materials-16-05945-t002:** Distances (r) between Ca and Mg atoms in the Ca2 position and O atoms around these atoms (distances up to 7 nearest O atoms) as the number of Mg atoms increases upon the substitution nMg/Ca2. The absolute and relative changes in these distances are given.

Atom Numbers		r(O–Ca2) (Å)							
Mg/Ca2	O	r_0_, 0 Mg/Ca2, x = 0	r_8Mg_, 8 Mg/Ca2, x = 1	∆r = r_8Mg_ − r_0_	∆r/r_0_ (%)	r_12Mg_, 12 Mg/Ca2, x = 1.5	r_16Mg_, 16 Mg/Ca2, x = 2	∆r = r_16Mg_ − r_0_	∆r/r_0_ (%)
Left OH channel									
#66	#154	2.80592	**2.53536**	**0.27056**	**9.6**	**2.53676**	**2.45072**	**0.3552**	**12.7**
#66	#178	2.28457	** *2.04952* **	** *0.23505* **	** *10.3* **	** *2.07599* **	** *2.10564* **	** *0.17893* **	** *7.8* **
#66	#233	2.30274	** *2.14333* **	** *0.15941* **	** *6.9* **	** *2.16441* **	** *2.18120* **	** *0.12154* **	** *5.3* **
#66	#258	2.37067	**2.24454**	**0.12613**	**5.3**	**2.25643**	** *2.19548* **	** *0.17519* **	** *7.4* **
#66	#274	2.31140	2.31617	−0.00477	−0.2	2.24902	2.24248	0.06892	3.0
#66	#314	2.43955	2.35683	0.08272	3.4	2.33322	2.40179	0.03776	1.5
#66	#332	2.31858	** *2.08125* **	** *0.23733* **	** *10.24* **	** *2.08557* **	** *2.08224* **	** *0.23634* **	** *10.2* **
Right OH channel									
#46	#328	2.42792	2.30753	0.12039	4.96	2.39495	2.38934	0.03858	1.59
#64	#336	2.41806	2.37853	0.03953	1.63	2.40006	2.38782	0.03024	1.25
#80	#328	2.31144	** *2.29482* **	** *0.01662* **	*0.72*	** *2.08659* **	** *2.08209* **	** *0.22935* **	** *9.92* **
#40	#336	2.42764	2.3995	0.02814	1.17	2.39543	2.37823	0.04941	2.04
#54	#328	2.41897	2.32563	0.09334	3.86	2.39074	2.38836	0.03061	1.27
#70	#336	2.30902	** *2.10253* **	** *0.20649* **	** *8.94* **	** *2.09402* **	** *2.0885* **	** *0.22052* **	** *9.55* **

Note: In bold italic font, the formation of the Mg–O ionic bond is marked. Cases of more significant changes in distances are simply marked in bold font.

**Table 3 materials-16-05945-t003:** The correspondence between the experimentally observed and modeled vibration bands.

Type of Vibration	Wavenumber (cm^−1^)				
	Experiment		Modeling		
	x = 0	x = 1	x = 0	x = 1 (Ca1)	x = 1 (Ca2)
O–P–O, ν_2_	Not measured	Not measured	425	425	425
O–P–O, ν_4_	570, 603	570, 603	520, 560	520, 555	520, 560
OH, ν_L_	630	630	640	620, 700	660, 700, 765
P–O, ν_1_	962	963	880	870	880
P–O, ν_3_	1048, 1088	1048, 1092	950, 1000	940, 1000	965, 1015
O–H, ν_S_	3573	3572	3640	3640	3620, 3655

**Table 4 materials-16-05945-t004:** Energy of formation E_f_ for Mg/Ca substitution in the Ca1 and Ca2 positions.

Mg Content		E_f_/(nMg/Ca1), eV/f.u.	E_f_/(nMg/Ca2), eV/f.u.	E_f_/(nMg/Ca1-OH), eV/f.u.	E_f_/(nMg/Ca2-OH), eV/f.u.
*n*	x				
0	0	0	0	0	0
1	0.125	0.11383	0.10233	-	-
2	0.25	0.10961	0.09742	-	-
4	0.5	0.1051	***0.09624*** *	0.10165	0.09651
8	1	0.10128	0.10285	-	-
12	1.5	0.10592	0.10124	-	-
16	2	0.11545	0.10210	0.11078	0.10

* Bold italic indicates the energy minimum at Ca2.

## Data Availability

The data presented in this study are available on request from the corresponding author.

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
