# Peer review of "Effect of Magnesium Substitution on Structural Features and Properties of Hydroxyapatite"

_materials, 2023, doi:10.3390/ma16175945_

Round 1
Reviewer 1 Report
To enhance the paper further, consider the following suggestions:
1. Add a specific research objective: Consider incorporating a specific research objective or question into the title. For example, "Investigating the Effects of Mg Substitution on the Structural Features and Properties of Hydroxyapatite" provides a clearer indication of the purpose and focus of the study.
2. Indicate the significance or potential applications: If applicable, consider adding information about the potential significance or applications of the research. For instance, "Enhancing Biocompatibility and Mechanical Strength: Structural Features and Properties of Mg-Substituted Hydroxyapatite for Bone Tissue Engineering."
3. Include quantitative results: Although the abstract mentions that changes in various parameters and properties were calculated, it would be beneficial to include some quantitative values or ranges to provide a clearer sense of the magnitude of these changes.
4. Emphasize the significance and implications: Highlight the potential significance and implications of the findings. For example, discussing how the observed changes in HAP-Mg properties and characteristics could impact its suitability for specific medical applications or improve our understanding of bone and tooth regeneration processes.
5. Clarify the limitations: It would be helpful to briefly mention any limitations or potential sources of error in the study, such as assumptions made in the DFT calculations or the sample sizes used in the experimental characterization.
6. Consider restructuring for better flow: The abstract jumps between different study aspects, including subsections or a more structured flow could enhance readability.
7. Provide more specific references: When referring to previous studies, it would be helpful to provide specific references or key studies exploring the effects of Mg substitutions in HAP. This would allow readers to delve deeper into the existing literature and establish a clearer context.
8. Clarify the research gap: Emphasize the specific aspects or contradictions in previous studies that need to be addressed in this research. This would help highlight the unique contributions of the current study and why it is necessary to investigate Mg-substituted HAP further.
9. Streamline repetitive information: The introduction contains some repetitive information and duplication of sentences. Streamlining the content and avoiding unnecessary repetition would improve clarity and readability.
10. Reiterate the significance of the findings: Emphasize the results' broader significance and potential impact on biomaterials and medical applications. Highlight how the detailed understanding of Mg/Ca substitutions in HAP can lead to improved materials and advancements in implant technology.
11. Address limitations and future directions: Discuss any limitations or challenges encountered during the study and propose potential future research directions. This could include investigating the specific effects of Mg/Ca substitutions on the interactions between HAP-Mg materials and living bone tissue and exploring the practical applications of the discovered electronic and vibrational changes.
12. Condense repetitive information: Some sentences in the conclusion appear repetitive, and condensing them would enhance the clarity and conciseness of the section.
Overall, the title is concise, relevant, and effectively conveys the main subject of the study. With minor improvements to incorporate a specific research objective or highlight the significance of the research, the title can become even more engaging and informative. The introduction provides a comprehensive overview of the importance and applications of HAP while highlighting the need for further research on Mg-substituted HAP. With the suggested improvements, the introduction can be more focused and engaging, setting a solid foundation for the subsequent study. The conclusion effectively summarizes the findings of Mg/Ca substitutions in HAP and highlights their implications for technological advancements and biomedical applications. With the suggested improvements, the conclusion can provide a strong ending to the study, reinforcing the significance of the results and suggesting potential avenues for future research.

Reviewer 2 Report
The current report uses DFT calculations to investigate the structural properties of magnesium-substituted hydroxyapatite. The Mg replacing HAp has been extensively studied in the literature, and the physical-chemical features of the materials have been highlighted (A. Bigi et al). The work in this study is well-done and yields substantial results, but there is no novelty regarding the medical application of the materials.
- The English language in general should be revised. There are long sentences, and the report should be edited by a native English speaker.
Author Response
Please see tha attachment

Reviewer 3 Report
In this manuscript, the authors used both experimental and DFT calculations to study the effects of Mg substitutions on HAP (Hydroxyapatite). The contents of this work are potentially interesting. However, the manuscript has various issues with the writing style including repetition, grammatical errors, incoherence, and a lack of conciseness, making it difficult to read due to poor English, and excessive unnecessary details. The results are also poorly presented and organized. Therefore, the manuscript should be major revised to address the following issues:
1) The introduction must be rewritten considering the following suggestions:
a. The author must avoid using long sentences like the ones on lines 18-22, 53-58, 76-80, 123-127, 133-137, 192-196. This is also true in other sections of the manuscript. For example, the sentences on lines 231-238, 240-244, 294-297, 299-303, 379-382, 717-724, 731-737, 812-816, etc.
b. Many sentences are improperly written, wordy, and repetitive. For example, these sentences “One of them is hydroxyapatite (Ca10(PO4)6(OH)2, HAP) [7–16], which is traditionally used in medicine. Due to its diverse properties and recent modifications, HAP, as a multifunctional material, is increasingly being used in other areas: environmental purification [17], photocatalysis [18] and photoluminescence [19]. It is also used as a material for cell imaging and drug release monitoring [20] and as a means of targeted drug delivery by HAP nanoparticles [21, 22]. Besides magnetic HAP nanoparticles (with iron inclusions) [23] are used for magneto-resonance imaging [24] and for local magnetic hyperthermia in the treatment of cancer tumors [ 23-25].” can rewrite as “One of them is hydroxyapatite (Ca10(PO4)6(OH)2, HAP) [7–16], which is traditionally used in medicine and other areas including environmental purification, photocatalysis, photoluminescence, cell imaging, drug delivery, cancer treatment, etc.”
Sentence on lines 183-185 “It should be noted that various theoretical numerical methods, modeling and calculations, including modern methods and DFT approaches [64, 84–87], have recently been increasingly used in the study of the effect of magnesium in HAP.” can change to “Advance computational approaches such as DFT calculations have been increasingly used to study the effect of magnesium in HAP [64, 84–87]”.
There are numerous instances of repeated sentences such as the sentence on lines 117-120 with the one on lines 115-117 as well as the one on line 138 with line 142. This is also true across the manuscript.
c. The following sentences should be rewritten because they are not clear: the one on lines 100-104, 138, and 156-158.
2) Authors must avoid using an excessive number of citations to support well-known topics. For example, on lines 36 and 37, the authors used 16 references, instead please cite only 4 references. This is also true on lines 67, 92, 104, 127, 161, 169, 254, and 616.
3) Trim unnecessary details. First, Section 2.2 should be updated because it has a lot of repetition and unnecessary information (I recommend describing this section on two pages rather than 3 ½ pages). Second, Section 3.2 should be rewritten to use only three pages, including figures, rather than nine. Avoiding using the phase such as “As can be seen, It should be noted that, It can be seen that, It is interesting that, Possible reasons for such differences in the behaviour of the unit cell parameters will be discussed below, The reasons for these features in the behavior of the HAP-Mg cell parameters will be discussed in more detail below, This observation will be discussed later, etc.” Furthermore, rather than focusing on details, summarize key points.
4) Combine or send some figures to Supplementary materials. The good paper contains no more than 6-7 figures. For example, figures 1, 2, and 3 can combine into one figure with a little modification. Figures 5 and 6 can also combine into one figure as well as figures 7 and 8. Figures 9 to 14 can combine into two figures.
5) Rewrite the conclusion in half a page rather than two pages.
Minor comments:
1) Modify Keywords by removing “modeling, hybrid functionals, substitution, nanomaterials, and properties” and adding more appropriate keywords.
2) Delete the following:
-Line 34, “however”
-Lines 53, 71, 83, 142, 163, 418, 543, 581, 594, 966, 971, 975, 987, 1012, 1015, and 1028 “At the same time”
-Lines 76, “therefore”
-Lines 82, 100, 110, 121, “here”
-Line 81, “It should be noted here that, in general”
-Line 171, “in order to evaluate the effect of Mg content on the properties of HAP”.
-Line 181, “as computational approaches, which makes it possible”.
-Line 188, “both”
-Line 189, replace “The inconsistency and contradictions in these previous studies call for new and more detailed studies here in order to ….” by “Therefore, further extensive studies are required to ….”
-Line 196, “.” after “positions”.
-Line 218, change “furrier” to “Fourier”.
-Line 227, “Technology and calculation algorithms.”
-Lines 372, 374, 391, and 396, change “HA” to “HAP”.
-Line 579, delete “(“.
-Line 1025, change “C1” to “Ca1”.
3) Please use the word "Figure" consistently; use either "Figure X" or "Fig. X," not both.
4) The atom colors in Fig.1(a) and Fig.1(b) are different, please make both figures have the same atom colors.
5) The distance labels on Figures 2, 9, 10, 11, and 13 are not clear, increase the font size.
6) Correct all typos.
There are many issues with the writing style including repetition, grammatical errors, incoherence, and a lack of conciseness, making the manuscript is so difficult to read due to poor English, and excessive unnecessary details.
Author Response
Plese see the attachment

Reviewer 4 Report
1. It should be provide the SEM image for the sample.
2. The conclusion of the paper is written like a summary of the work. Authors should rewrite this part concisely adding some perspective sentences.
3. Make sure all abbreviations are written out in full the first time used. This is particularly important in the abstract and the conclusions but work through the entire ms carefully from this perspective.
4. Some refs on Methodological features. Technology and calculation algorithms, such as
Theor. Chem. Acc. 2022, 141, 68; J Comput Chem, 2018, 39, 117–129; ACS Omega, 2018, 3, 17986−17990; J. Phys. Chem. A, 2019, 123, 6751−6760 and Chemical Physics Letters, 2015, 633, 265–272
5. At certain places language is awkward. So, please check the minor inconsistencies.
6. Give the PXRD index.
minor revision
Round 2
Reviewer 4 Report
accept